

# SubVis: an interactive R package for exploring the effects of multiple substitution matrices on pairwise sequence alignment

Scott Barlowe[1], Heather B. Coan[2] and Robert T. Youker[2]

[1] Department of Mathematics and Computer Science, Western Carolina University, Cullowhee, NC, United States of America
[2] Department of Biology, Western Carolina University, Cullowhee, NC, United States of America

## ABSTRACT

Understanding how proteins mutate is critical to solving a host of biological problems. Mutations occur when an amino acid is substituted for another in a protein sequence. The set of likelihoods for amino acid substitutions is stored in a matrix and input to alignment algorithms. The quality of the resulting alignment is used to assess the similarity of two or more sequences and can vary according to assumptions modeled by the substitution matrix. Substitution strategies with minor parameter variations are often grouped together in families. For example, the BLOSUM and PAM matrix families are commonly used because they provide a standard, predefined way of modeling substitutions. However, researchers often do not know if a given matrix family or any individual matrix within a family is the most suitable. Furthermore, predefined matrix families may inaccurately reflect a particular hypothesis that a researcher wishes to model or otherwise result in unsatisfactory alignments. In these cases, the ability to compare the effects of one or more custom matrices may be needed. This laborious process is often performed manually because the ability to simultaneously load multiple matrices and then compare their effects on alignments is not readily available in current software tools. This paper presents SubVis, an interactive R package for loading and applying multiple substitution matrices to pairwise alignments. Users can simultaneously explore alignments resulting from multiple predefined and custom substitution matrices. SubVis utilizes several of the alignment functions found in R, a common language among protein scientists. Functions are tied together with the Shiny platform which allows the modification of input parameters. Information regarding alignment quality and individual amino acid substitutions is displayed with the JavaScript language which provides interactive visualizations for revealing both high-level and low-level alignment information.

Corresponding author
Scott Barlowe,
sabarlowe@email.wcu.edu

## INTRODUCTION

Prediction of protein similarity through sequence alignment is an important tool for a number of biological applications including the understanding of evolutionary divergence,

identification of active/conserved regions in proteins, and identification of key structural motifs in proteins. Identification of similarities among protein families, individual proteins, or even short segments of a protein chain can give scientists insights as to how an amino acid insertion or mutation may alter the active regions within the putative protein. Accurate alignment of two or more proteins being compared is an important first step in evaluating similarity and many algorithms exist that use a wide range of criteria to find the best alignment (*Ma & Wang, 2014*; *Haque, Aravind & Reddy, 2009*; *Gotoh, 1999*; *Li & Homer, 2010*).

Alignments are highly dependent on algorithm parameters, such as gap penalties and scoring type (local or global). One of the parameters influencing alignment scores is the chosen substitution matrix capturing the likelihood of amino acid substitutions (*Altschul, 1991*). Substitution matrices capture the likelihood of amino acid substitutions by reporting the log-odds ratio of each possible substitution calculated by

$$s_{ij} = \frac{\ln \frac{q_{ij}}{p_i p_j}}{\lambda} \tag{1}$$

where, for amino acid $i$ and amino acid $j$, $s$ is the substitution score, $q$ is the set of observed frequencies, $p$ is the probability of random appearance, and $\lambda$ is a positive scaling constant allowing for the use of different logarithm bases without changing observed frequencies. Conservative substitutions have a positive score and non-conservative substitutions have a negative score (*Pearson, 2013*). Standard, predefined matrices offer a quick way to model substitutions and those matrices that differ only by variations of selected parameters can be grouped into families. Two well-known matrix families are the PAM (*Dayhoff, Schwartz & Orcutt, 1978*; *Schwartz & Dayhoff, 1978*) and BLOSUM (*Henikoff & Henikoff, 1992*) matrices. A summary of both matrix families is given by *Pearson (2013)*. Although both PAM and BLOSUM find the log-odds ratios for matrix values, each family has different methods to calculate the likelihood of substitution. PAM matrices are based on the mutation frequency of closely related proteins which is then extrapolated to more distant evolutionary lines. Instead of extrapolation of highly related proteins, BLOSUM matrices calculate frequencies by locating conserved blocks and then use a threshold to exclude closely and moderately related proteins (for a more detailed discussion on how PAM and BLOSUM matrices affect alignments we direct the reader to (*Mount, 2008*; *Altschul, 1991*; *Pearson, 2013*). Other matrix families exist (*Müller, Spang & Vingron, 2002*; *Benner, Cohen & Gonnet, 1994*) and the development and analysis of additional families is a subject of ongoing research.

Predefined matrices may not adequately model substitutions for a variety of reasons. New substitution strategies may be required and result in the modification of existing matrices (*Yu & Altschul, 2005*) or the construction of entirely new ones. Reasons why predefined matrices may not accurately model substitutions include the following scenarios: application-specific alignments (*States, Gish & Altschul, 1991*; *Paila, Kondam & Ranjan, 2008*), matrix optimization (*Saigo, Vert & Akutsu, 2006*), compensating for non-conventional amino acid composition (*Jimenez-Morales, Adamian & Liang, 2008*), aligning distantly related sequences (*Prlić, Domingues & Sippl, 2000*), accounting for site

specific dynamics in phylogenetic models (*Wang et al., 2008*), and incorporating structural information (*Vilim et al., 2004*; *Teodorescu et al., 2004*; *Goonesekere & Lee, 2008*), (for another survey of custom substitution matrices, see *Yamada & Tomii (2014)*). While alignment of sequences is central to phylogenetic studies, these investigations are commonly performed at the nucleotide or codon level. In the case of protein-level phylogenetic studies, specialized, standardized matrices are vital for assessing evolutionary heterogeneity (*Wang et al., 2008*; *Wang, Susko & Roger, 2014*; *Rokas & Carroll, 2008*; *Dean et al., 2002*; *Echave, Spielman & Wilke, 2016*). Examples of phylogenetic-specific matrices include JTT (*Jones, Taylor & Thornton, 1992*) and WAG (*Whelan & Goldman, 2001*). Despite the sophistication of phylogenetic models, these matrices were not built for use in protein alignment with customizable matrices and to visualize structure/function predictions (*Whelan & Goldman, 2001*; *Wang et al., 2008*; *Wang, Susko & Roger, 2014*).

Multiple substitution matrices can be compared to find the most appropriate one (*Altschul, 1991*). *Rios et al. (2015)* and *Agrawal & Huang (2009)* illustrate the importance of comparing pairwise alignments produced by varying substitution matrices. However, this can be a difficult task. Comparison often includes analysis of both alignment quality and behavior at individual amino acid positions. If using predefined matrices, it may not be known which matrix family most accurately reflects the likelihood of individual substitutions among the proteins being studied. Even within a family, one matrix may be more suitable than others given a specific application (*Altschul, 1991*). Furthermore, none of the predefined matrix families may adequately represent a scientist's knowledge about a particular set of proteins. In the latter case, custom matrices are required to achieve accurate alignments which often need to be compared to other widely used or custom matrices.

There are few tools for addressing the complex problem of choosing the most appropriate substitution matrix for protein sequence alignments. Because of these needs, we have developed SubVis, a highly interactive R (*R Core Team, 2013*) package that allows the simultaneous visual exploration of how varying substitution matrices affect alignment results. To address the shortcomings of previous tools, SubVis

- Allows the uploading or text entry of FASTA (*Pearson & Lipman, 1988*) sequences.
- Utilizes widely-known R functions from the Biostrings package (*Pages et al., 2016*).
- Permits the application of several widely-used substitution matrices and multiple custom matrices.
- Provides intuitive and interactive visualizations to facilitate simultaneous exploration of protein alignments produced by multiple substitution matrices. Detail information, such as the log-odds score for each substitution, is available through mouse interaction.
- Employs the Shiny package (*Chang et al., 2016*) and JavaScript for web-based parameter loading and visualization, respectively.

In this paper, we present background information including the difficulties associated with choosing a substitution matrix and previous attempts using visualization to help understand the effect of substitution matrices. We describe the organization and implementation of SubVis. Finally, we provide a case study illustrating the utility of SubVis.

## BACKGROUND

### Alignment quality

Performing quality pairwise sequence alignments is a critical first step in protein analyses such as the formation of multiple sequence alignments and phylogenic tree construction (*Agrawal & Huang, 2009*). As described in detail by *Landan & Graur (2008)*, alignments are subject to a host of errors, such as the lack of parameters accurately reflecting true conditions before analysis is performed. This lack of *a priori* information makes the seriousness of the error difficult to judge and contributes to uncertainty that obfuscates biological insight.

Summary statistics can be useful for eliminating poor alignments from analysis during the initial investigation. However, summary statistics can be problematic if they are not supplemented by detailed exploration. For example, percent identity is a simple, popular metric but suffers from several deficiencies, including high uncertainty and important calculation variations that are mostly ignored (*Raghava & Barton, 2006*). Another aggregate quality metric is the alignment score which accounts for substitution scores and gap penalties (*Henikoff, 1996*). However, many different alignments can result in the same score (*Landan & Graur, 2008*). Furthermore, scoring functions can be suboptimal and result in an alignment with a higher error being assigned a higher score. *Edgar & Sjölander (2004)* illustrate some of the problems associated with assigning scores by analyzing three quality measures. Each presented score has drawbacks that include not compensating for over-alignment, under-alignment, alignments offset from the reference alignment, and a scoring function that itself requires decisions regarding parameter input. Statistical significance represented by a *P*-value is often used to judge assigned alignment scores (*Mitrophanov & Borodovsky, 2006*). However, this descriptor can suffer from assumptions about the model of randomness used and from the fact that multiple *P*-value methods may be needed when varying either the alignment parameters or the alignment algorithm.

The choice of substitution matrix is critical to defining and producing a quality alignment. This choice becomes more important as alignment uncertainty increases (*Henikoff, 1996*). However, evaluating substitution matrices can be a difficult task. Complex relationships among variables that affect protein mutations are often simplified with model assumptions which may not be correct (*Crooks, Green & Brenner, 2005*). Furthermore, substitution matrix evaluation can depend on some of the same factors described above, including alignment scope (local or global) and whether gap penalties are applied (*Henikoff, 1996*). *Agrawal & Huang (2009)* illustrate the type of analysis that is made difficult with the range and variability of substitution matrix choice. Their work evaluates 15 substitution matrices with a range of parameter sets. For each of the matrices, several alignment quality measures are compared. Although substitution matrix evaluation is crucial in producing quality alignments, there is a shortage of tools that can accommodate variability in parameters, are able to scale to a large number of matrices, and that allow exploration beyond summary quality measures.

## Visual approaches

Interactive visualization can be useful when comparing alignments resulting from the application of multiple substitution matrices and varying parameter sets. Furthermore, a tool that allows visual exploration can help uncover the details hidden in summary statistics. However, there have been few approaches applying interactive visualization to the analysis of substitution matrices. *Bulka, desJardins & Freeland (2006)* extends the work of *Nakai, Kidera & Kanehisa (1988)* and the work of *Tomii & Kanehisa (1996)* to present a web-based tool. The tool uses a color-coded minimum spanning tree to visualize the similarities of amino acid indices to a substitution matrix. Although these provide insight into the substitution similarity, the visual display does not reflect the spatial context inherent in typical two- or three-dimensional alignment representations. Additionally, the tool is limited in the amount of detail available through interactions. *Eyal, Pietrokovski & Bahar (2007)* presents another web-based platform that performs multiple sequence alignment based on pair-to-pair substitution matrices. However, it is designed for a specific custom matrix type, does not provide standard matrices, and lacks features that facilitate comparison among different matrices. CRASP (*Afonnikov & Kolchanov, 2004*) is a web-accessible tool that takes protein family sequence alignments, a phylogenetic tree or other weights, physicochemical characteristics, and conservation filters as input. The output consists of a correlation matrix, hierarchical clustering diagram, positional frequency statistics, and physicochemical descriptors. Additional output includes statistical estimators of coordinated substitution contributions. Their approach is limited to cases where the substitutions are thought to be highly correlated with one another and using a matrix or proteins which do not reflect this assumption could lead to inaccurate alignments. Much of the output is visualized as text with limited interactions.

Despite these attempts, much of the comparison and other analysis is still either performed manually or with tools that lack the flexibility provided by combining standard substitution matrices, custom substitution matrices, visualization, and robust interactions. We are not aware of any tool explicitly designed for integrating alignment algorithms and interactive comparison across a range of substitution matrices.

SubVis addresses many of the limitations to currently available platforms. SubVis allows scientists to load a pair of proteins to be aligned, choose basic parameters (such as alignment score type and gap penalties), and apply multiple substitution matrices. Applied matrices can include PAM matrices, BLOSUM matrices, or custom matrices. After performing the alignment, options exist for the high-level exploration of percent identities and alignment pair scores. Interactions also exist for the low-level exploration of individual amino acids across selected substitution matrices by position in the aligned sequence, properties (hydrophobic, physicochemical properties, volume, conserved or not, etc.), pattern matching, and locations of insertions and deletions (indels).

## IMPLEMENTATION

SubVis consists of three functional units: interface and parameter management, alignment processing, and visualization. Interface and parameter management controls the capture

of alignment parameters, including selected substitution matrices. It also allows the user to choose the visualization (overview, detail view, or search view) and options for selecting and searching displayed items. Alignment processing accepts alignment parameters captured from the interface and includes those in the construction of alignments. The visualization component displays the alignments after each of the selected parameters (including the substitution matrix) has been applied and provides detailed information with mouse interaction.

## Interface and parameter management with Shiny

Shiny is a recently developed R package for building web applications and was chosen for this system because of the GUI widgets available and its integration with R. The first screen shown when starting SubVis is the parameter view under the "Options" tab which captures alignment input such as the proteins to be aligned, the predefined and custom matrices to be applied, gap penalties, and scoring type. Users are allowed to load (and change) the following parameters:

- **Protein sequences.** Two protein sequences in FASTA (*Pearson & Lipman, 1988*) format can be loaded by selecting the sequence file from the local computer or entering the sequences into text boxes manually, including with copy and paste. If sequences are entered into the text boxes, FASTA files are created in the package directory structure for future reference. One sequence represents the *pattern* and the other represents the *subject*. (Sequences are referred to as the *pattern* or *subject* to be consistent with the Biostrings package where they are defined in context of the functions utilized in the SubVis implementation.)
- **Predefined substitution matrices.** Multiple PAM and BLOSUM matrices can be selected by checking the corresponding boxes. Individual gap penalties can be entered for each predefined matrix. Predefined PAM matrices included in SubVis are PAM30, PAM40, PAM70, PAM120, and PAM250. Predefined BLOSUM matrices included in SubVis are BLOSUM45, BLOSUM50, BLOSUM62, BLOSUM80, and BLOSUM100.
- **Custom substitution matrices.** Multiple space delimited text files each containing a custom matrix can be loaded. Users can load custom multiple matrices by selecting a master file that lists the filename of each matrix. In the master file listing the custom matrices, each filename is on a separate line. Following each filename on the same line are space delimited gap penalties for each custom matrix. In addition to exploring different matrices, users can explore the effects of penalties by repeating the same matrix file name with variations in gap and extension penalties. Figure 1 shows the relationship between the master file and the custom matrices.
- **Alignment score type.** Users can choose from local, global, overlap, local–global, and global-local scoring.
- **View choice.** Clicking the "GO" button in the parameter capture view performs the alignment and automatically switches to the "VIZ" tab where users can choose from three visualization views. The overview provides quality information by sorting and displaying four percent identity variations (*May, 2004*; *Raghava & Barton, 2006*) and the overall alignment score. Based on this information, matrices can be excluded or included

Custom Matrix Files

File Listing Custom Matrices

| *Custom_Filename_0* | *Gap_Cost_0* | *Extension_Cost_0* |
| --- | --- | --- |
| *Custom_Filename_1* | *Gap_Cost_1* | *Extension_Cost_1* |
| ... | ... | ... |
| *Custom_Filename_N* | *Gap_Cost_N* | *Extension_Cost_N* |

**Figure 1  Loading custom matrices.** Multiple custom matrices can be loaded by creating a master file listing the filenames and penalties associated with each matrix. In the master file, each line consists of the filename followed by the gap and extension penalties associated with individual matrices. Specific requirements for formatting the custom matrix master file can be found in the help contents. Several example custom matrices and master files are included in the software package.

in the detail view. The detail view shows individual amino acids as either color-coded boxes or the single letter abbreviation, the classification of amino acid properties, and the log-odds score for each substitution. This view also allows alignment navigation. The search view allows searching by amino acid position in the aligned sequence, matching sections in the alignment pair, indel location, and subsequence matching. The overview, detail view, and search view can provide information that aids in the analysis of which substitution model is the most suitable for a given scenario. Users can change views simply by clicking on the desired tab or selecting the appropriate visualization from a drop-down menu. Features available for each view are listed in Table 1 and will be discussed in detail later.

## Sequence processing with R

After capturing input with Shiny, SubVis reports the parameters to the alignment processing component and utilizes functions from the Biostrings package to perform sequence alignment, calculate alignment scores, capture indel locations, perform any other necessary alignment/string manipulations, and communicates input changes to the visualization component. The primary functions used by SubVis are described below (more detailed information can be found in the Biostrings documentation):

- **pairwiseAlignment.** Accepts the two protein sequences (*pattern* and *subject*), gap costs, alignment score type, and substitution matrices entered as parameters. Alignment choices include local alignments using the Smith–Waterman algorithm (*Smith & Waterman, 1981*), global alignments with the Needleman–Wunsch algorithm (*Needleman & Wunsch, 1970*), and overlap algorithms with an ends free algorithm. There are also two mixed scoring types: local–global scoring and global-local scoring. All other parameters are left at default values. SubVis invokes this function once for each substitution matrix. Because SubVis is open source, users can implement other alignment algorithms or substitute alignments produced by other tools.

**Table 1 Views available in SubVis.** Beside each view is a list of interactions and information available for capturing parameters and visualization (overview, detail view, and search view).

| Parameter capture | Overview |
| --- | --- |
| Input protein sequences | Matrices sorted by percent identity |
| Select predefined matrices | Matrices sorted by overall alignment score |
| Load custom matrices | Individual matrix scores |
| Input penalties per matrix | Individual matrix percent identity |
| Select scoring type | |

| Detail view | Search view |
| --- | --- |
| Pairwise alignments per matrix | Search by amino acid position |
| Amino acid names and positions | Search for indels |
| Amino acid substitution scores | Search for matches in alignment pairs |
| Multiple amino acid classifications | Search for input sequences |
| One letter amino acid abbreviations | |
| Alignment navigation | |
| Subject/pattern filtering | |

- **matchPattern.** Finds all occurrences of an input pattern. The output is the starting and ending points of matches.
- **indel.** Finds gaps in the alignment resulting from insertions and deletions in the aligned sequences.
- **pid.** Calculates four percent identity types as reported by *May (2004)* and evaluated by *Raghava & Barton (2006)* where differences in denominator calculation reflect variations in defining sequence length. Parameters to this function indicate if the denominator should be defined as aligned positions plus internal gap positions (PID 1), aligned positions (PID 2), the length of the shorter sequence (PID 3), or the average of the two sequences (PID 4). For each selected matrix, SubVis sorts and then displays the four unique percent identities in a color-coded row.

Before parameters are passed to alignment functions, they are checked for values and formats that may cause system errors. Tailored error messages include those for missing sequence files, missing penalties, and identical sequences. An error is also produced if the custom matrix option is enabled but a file listing the matrices has not been selected. SubVis generates a general error message if the *pairwiseAlignment* function defined by the Biostrings package fails. Possible causes of this error are poorly constructed sequences or custom matrices.

## Visualization with JavaScript

After constructing alignments based on user input, the alignments and supporting information are displayed. The interactive visualization component consists of an overview, a detail view, and a search view developed in JavaScript with information passed to it from R.

## Overview

Despite the problems associated with summary statistics, they can be useful in preliminary analysis to help narrow the number of alignments being explored in detail. Percent identity is a commonly used measure in sequence alignment but variations in how it is calculated are not typically reported even though these differences can affect alignment assessment (*Raghava & Barton, 2006*). The overview in SubVis provides a high-level perspective of the alignments by sorting and then displaying the four variations of percent identities provided by the *pid* function in the Biostrings package. The sorted percent identities for each matrix type are shown in a color-coded bar (Fig. 2A) normalized and colored from blue (lowest percent identity) to red (highest percent identity). Under the set of PID rows is a normalized, color-coded bar of all matrix types sorted by alignment score. A legend in the bottom-left corner illustrates how colors correspond to the PID and score range. When the mouse moves over either a matrix's percent identity or alignment score, the same matrix is selected in the other rows (Fig. 2B) to ease comparison. At the same time, the numerical value of the percent identity and alignment score are displayed in the lower right corner. After exploring the overview, substitution matrices can be removed or added by revisiting the "Options" tab.

## Detail view

After investigating alignments based on percent identities and alignment score, alignments and individual amino acids can be explored in the detail view (Fig. 3). If the mouse is not over an amino acid, only basic information such as protein chain names, the matrix type per alignment pair, the score per alignment pair, and the amino acid position range after alignment is displayed. By default, amino acids are represented by colored boxes where an amino acid corresponds to a single color and gaps are black.

If the mouse moves over a single amino acid, additional details appear. In the top-right corner, additional information includes the log-odds score, the substitution that occurred, the name of the selected amino acid, and the aligned position. (For amino acid—gap pairs, SubVis reports the log-odds score as undefined.) In the top-left corner, the gap penalties for that alignment are displayed along with the selected score type. Beneath the alignment score for each matrix type along the left side, the log-odds score and the substitution that occurred are displayed for amino acids appearing in the same column as the one selected. Above the set of alignments is a histogram that shows the type and number of the amino acids (and gaps) occurring in that column.

Classifying amino acids according to their properties is an important part of protein research (*Biro, 2006*; *Koshi & Goldstein, 1997*; *Pommié et al., 2004*; *Bulka, desJardins & Freeland, 2006*; *Aftabuddin & Kundu, 2007*). SubVis allows amino acids in the aligned sequences to be classified into groups based on the physical and chemical properties of interest by selecting that group from a drop-down box (Fig. 4). Groups are color-coded where a color corresponds to a single group. This simplifies alignment analysis by allowing groups of amino acids sharing common characteristics to be compared instead of individual amino acids. We use the classification scheme presented by *Pommié et al. (2004)*. Table 2 shows the classes and subgroups. A legend of the grouping is shown at the top of the

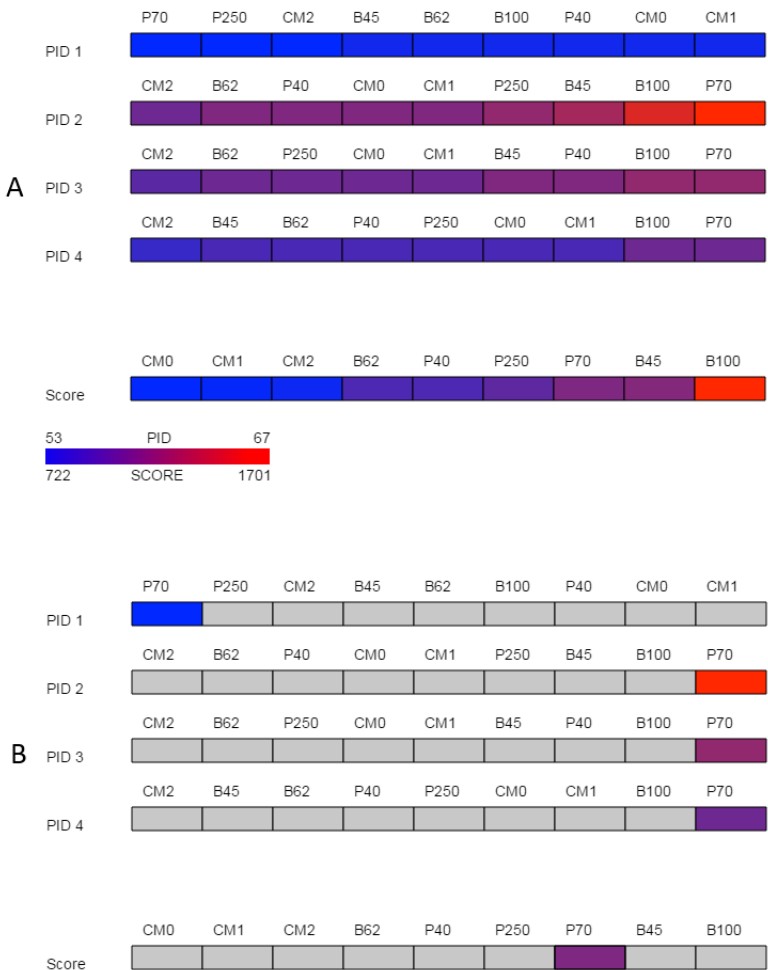

**Figure 2  Overview.** The overview provided by SubVis allows investigation of four unique percent identity calculations and the alignment score per substitution matrix. The display before interaction is shown in (A) and the highlighted substitution matrix selected with mouse movement is shown in (B). Custom matrices begin with the prefix "CM" followed by their position in the master file. A legend in the bottom-left corner lists how the maximum and minimum alignment score and PID correspond to color. In this example, the PAM70 matrix has a relatively high alignment score and PID except for PID 1 for which PAM70 is the lowest. The sequences used for this figure are *G-protein coupled receptor 6 isoform b* and *G-protein coupled receptor 12* from the rhodopsin family analyzed by *Fredriksson et al. (2003)*. The custom matrix file lists a single matrix with varying, user-defined penalties and was developed for studying the transmembrane region of G protein-coupled receptors from the rhodopsin family (*Rios et al., 2015*). The same sequences and parameters are used in Figs. 3 and 4.

display and the histogram is also colored by group. Additionally, substitution pairs can be grouped as conservative (log-odds score $> 0$) or non-conservative (log-odds score $< 0$) (*Pearson, 2013*).

There are many additional interactions to ease alignment navigation. Instead of colored boxes, the single letter amino acid abbreviation can be displayed. The default layout shows both the *pattern* and *subject* for each pair. Alignment sequences can be navigated forward and backward by clicking a button. To maintain positional context, incrementally moving forward or backward only shifts the alignment one-half of the number of amino

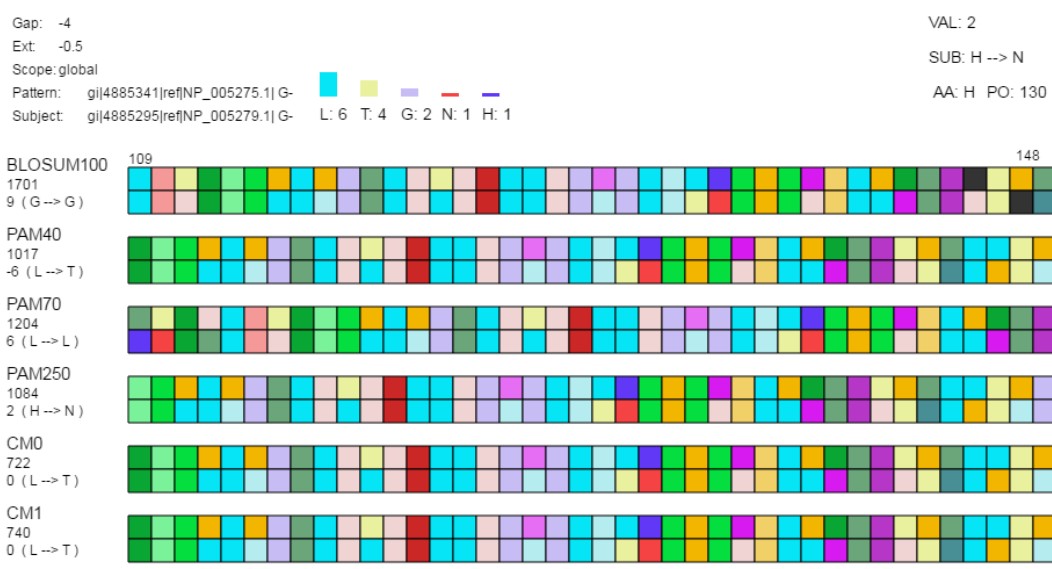

**Figure 3 Detail view.** The detail view allows investigation of individual amino acids. Aligned sequences are shown as *pattern-subject* pairs in the center using color-coded boxes (as shown) or one letter abbreviations to represent individual amino acids. When the mouse moves over an individual amino acid (1) the specific amino acid substitution occurring for that position in the aligned sequences and the corresponding log-odds score are shown under the alignment score for each pair along the left side of the display; (2) the top-left displays the gap cost, extension cost, and the score type; (3) a histogram is shown above the set of alignment pairs displaying the frequency of each amino acid in the selected column for all pairs; and (4) the log-odds score ("VAL"), the specific amino acid substitution ("SUB"), the current amino acid ("AA"), and the position in the aligned sequence ("PO") are displayed in the top-right.

acids currently displayed. SubVis also has an option for showing only the *pattern* or only the *subject*.

### Search view

The search view includes several options for locating a desired alignment region. Users can search alignments by amino acid position in the aligned sequence by entering the position number into a text box. Searches can also locate indels, regions where the pattern and subject of an alignment pair match, and sections that match an input sequence. Indel locations and areas that fulfill match criteria are shown as red with the remainder of the sequence in gray.

## CASE STUDY

We now present an example of how SubVis can aid in the exploration of alignment sequences produced by multiple matrices and their associated penalties. Intrinsically disordered proteins (*Wright & Dyson, 1999*; *Dunker et al., 2002*; *Dyson & Wright, 2005*) contain functional regions associated with ill-defined fold structures. Although intrinsically disordered proteins are thought to participate in important functions such as network signaling and regulation, their lack of a stable, predictable structure makes designing effective analysis tools difficult. For example, *Radivojac et al. (2002)* attempted to construct a substitution matrix for disordered proteins. They tested their matrix, called DISORDER,

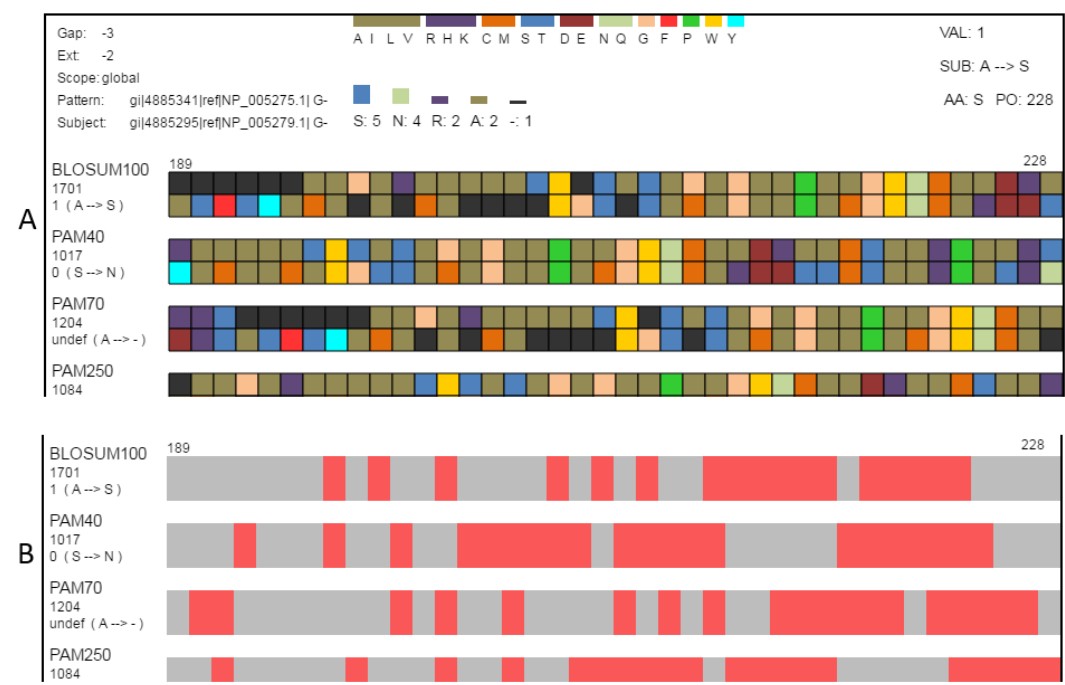

**Figure 4   Classification and searching.** (A) Amino acids can be grouped according to the seven classifications (Table 2) reported by *Pommié et al. (2004)*. When a classification is selected, a legend showing how amino acid colors correspond to classification groups is displayed in the top-center and the histogram is recolored to match subgroups. Amino acids can also be grouped as conservative or non-conservative. The physicochemical classification is shown here. (B) The same region in the search view where matches to one of the search criteria are colored in red. This figure shows locations in the view where the alignment pairs match.

**Table 2   Amino acid classification groups per the scheme found in *Pommié et al. (2004)*.**

| Hydropathy | Volume | Chemical | Charge | Hydrogen Don/Acc | Polarity | Physicochemical |
|---|---|---|---|---|---|---|
| Hydrophobic | Very small | Aliphatic | Positive | Donor | Polar | Aliphatic |
| Neutral | Small | Aromatic | Negative | Acceptor | Nonpolar | Basic |
| Hydrophilic | Medium | Sulfur | Uncharged | Both | | Sulfur |
| | Large | Hydroxyl | | None | | Hydroxyl |
| | Very large | Basic | | | | Acidic |
| | | Acidic | | | | Amide |
| | | Amide | | | | G |
| | | | | | | F |
| | | | | | | P |
| | | | | | | W |
| | | | | | | Y |

on a wide range of disordered proteins and found that it did not produce a notable cumulative score improvement over BLOSUM62.

*Radivojac et al. (2002)* notes that evaluating the performance of new matrices is more difficult than their construction. SubVis allows detailed exploration of the effect of substitution matrices and eases the alignment analysis for a given set of proteins, especially for insights about specific regions that may be hidden in aggregate scores. For example, we desired to test the hypothesis that the DISORDER matrix would perform better than the BLOSUM62 matrix when applied to the disordered DDX4 human and *Xenopus laevis* proteins. The BLOSUM62 and DISORDER matrices were both loaded into SubVis as custom matrices using a subset of the associated gap/extension costs reported by Radivojac et al. (Note that although the entire set of matrices could have been loaded for any pair of proteins, a smaller set makes our example more concise.) Specifically, the subset included the following custom matrices listed by label and matrix type followed by *(gap cost, extension cost)*:

**CM0:** DISORDER $(-3.2, -0.1)$
**CM1:** DISORDER $(-10, -0.6)$
**CM2:** DISORDER $(-7.5, -0.9)$
**CM3:** BLOSUM62 $(-3.2, -0.1)$
**CM4:** BLOSUM62 $(-10, -0.6)$
**CM5:** BLOSUM62 $(-7.5, -0.9)$

The overview produced by SubVis (Fig. 5A) shows that the BLOSUM62 matrix generally performs better. For instance, the top three alignment scores are BLOSUM62 matrices and the bottom three alignment scores are DISORDER (*Radivojac et al., 2002*) matrices. Furthermore, all percent identities except for PID 2 have BLOSUM62 matrices as three out of the four highest percent identities. Evident from the color distribution, PID 2 results in the highest percent identities (72 max, 64 min) but has a similar ordering as the others except for a shuffling at the lower end. The two consistently best PID performers are DISORDER (CM0) and BLOSUM62 (CM3), both of which were produced with gap and extension costs of $-3.2$ and $-0.1$, respectively. For these two matrices, the maximum difference across all PID types is only one percent. Because the PID for CM0 and CM3 are similar but their alignment scores are less similar, we decided to explore those alignments in more detail.

The search view and detail view in SubVis allowed us to learn more about the similarities and differences between CM0 and CM3. In the search view, individual regions were visually scanned by incrementally advancing the alignments from beginning to end. The region outlined with solid black rectangles in Fig. 5B shows aligned regions that have similar match patterns in CM0 and CM3. Examining the single letter abbreviations in the detail view shows that the alignments are identical except for a single column offset (Fig. 5C). The percent identity is the same for both regions but we wanted to find more detail about the substitution scores. Simple mouse interaction in SubVis allowed us to find where there are substitution scores in that region that differ between matrices. For example, CM0 scores the substitution of glutamic acid (E) to glutamine (Q) as 0. However, CM3 scores this substitution as 2 (Fig. 5C). Classifying the properties of the amino acids indicates

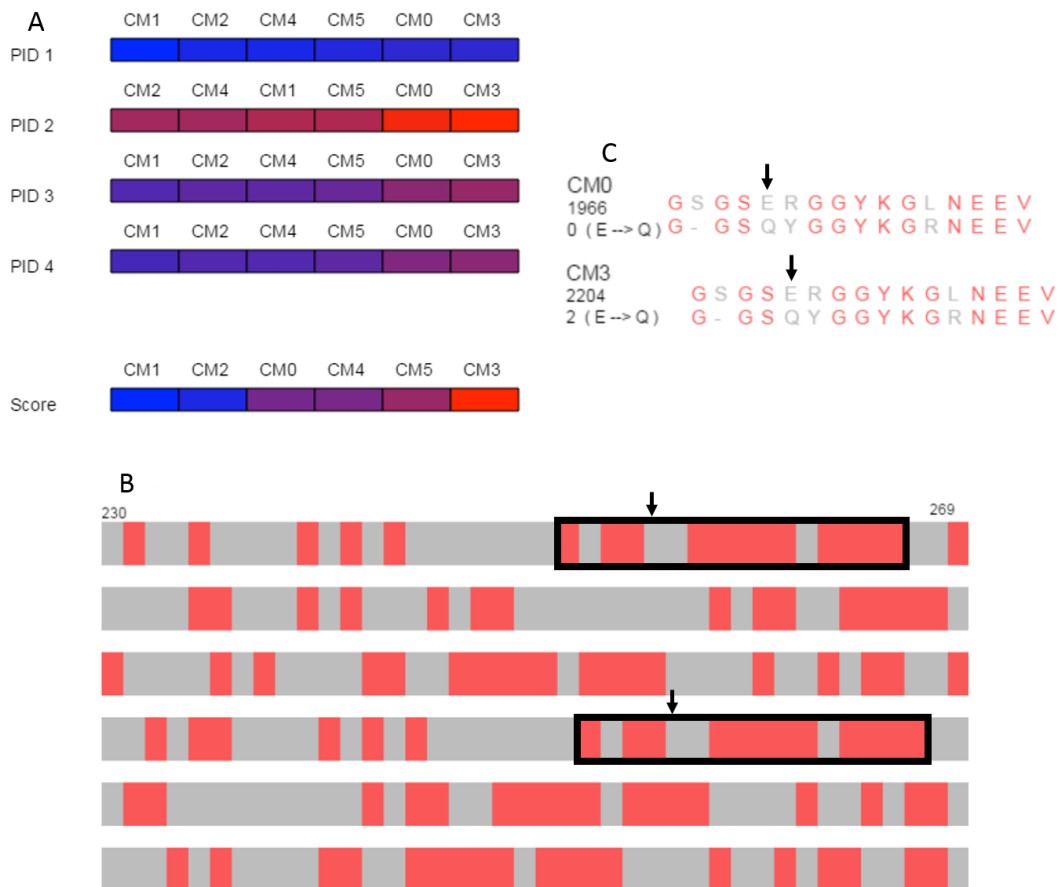

**Figure 5** **Case study.** (A) Overview of PID and alignment score calculations show that BLOSUM62 generally outperforms DISORDER (*Radivojac et al., 2002*) for the DDX4 human and *Xenopus laevis* proteins. CM0 (DISORDER) and CM3 (BLOSUM62) have similar penalties and PID results but relatively different alignment scores. (B) Browsing the alignments in the search view shows similar patterns for CM0 and CM3 that are marked with black rectangles in the figure. Black arrows indicate the location of the glutamic acid to glutamine substitution. (C) The single letter amino acid abbreviation with the substitution and substitution score that appear when the mouse moves over the amino acid pair marked with arrows. The figure above was produced with a local alignment but a global alignment produced similar results for both the overview and for the outlined regions.

why this substitution score is low for both matrices by showing that they share the same group for only hydropathy, volume, and polarity. Furthermore, CM3 has substitution scores that are greater than or equal to the corresponding substitution in CM0, except for the single tyrosine (Y) match. In that case, the substitution has a higher value in CM0. In cases where the region of interest is longer and expands across a larger, more varied range of substitution matrices, manually comparing the substitution values for even a limited set of substitutions can become cumbersome. SubVis can aid analysis even in these more complex cases by making the classification of amino acids and the score for a substitution quickly available.

## RESULTS AND DISCUSSION

SubVis allows scientists to load protein sequences and visually explore alignment differences that result from varying predefined and custom substitution matrices. This platform allows scientists to view coarse-grain and fine-grain information ranging from summary alignment scoring to specific amino acid substitution details. Additional interactions include searching multiple alignment pairs and classifying amino acids according to a selected property. The ability to load sequences, apply desired alignment parameters (including substitution matrices), search alignments, classify amino acids, and access detailed substitution scores facilitate the comparison of established substitution matrices and the evaluation of matrices being developed for specific purposes.

SubVis utilizes general R programming constructs and the Biostrings package, both of which are well-known in the bioinformatics community. SubVis is available as an R package on CRAN. The package contains the data sets and custom matrices used to produce the presented figures. There is also a detailed vignette included in the package and demonstration videos located on GitHub. The SubVis package allows users to access the vignette through a "Help" tab persistent in all views. The help content includes (but is not limited to) descriptions of interactions, specific error messages, and the specific format of the file listing custom matrices and their associated penalties. The help section also explains the location of created files (sequence files created by text box entry and custom matrices) and when read/write permissions for those locations may be needed. The help content is organized by subject and can be accessed quickly by clicking on corresponding links at the top of the page.

## CONCLUSIONS AND FUTURE WORK

Substitution matrices are crucial to alignment algorithms but current tools do not allow the simultaneous exploration of alignments resulting from multiple matrices. This work presents SubVis, an interactive R package for visually exploring the effects of substitution matrices on protein sequence alignment. Widely used matrices and multiple user-defined custom matrices can be applied to alignments. SubVis utilizes Shiny for capturing parameters, R to process alignments, and JavaScript to visualize overview and detail results. Users can easily transition from overall metrics, such as percent identity and alignment score, to detailed information for individual amino acids and vice versa. Many interactions allow the display of desired information including log-odds ratios, pattern matches, and amino acid classification by property.

There are many opportunities for future work. For example, we plan to extend SubVis from pairwise sequence alignments to multiple sequence alignments and include more descriptors of alignment quality. We would also like to include the ability to dynamically build or modify individual substitution matrices and then immediately investigate the effects of changes on the alignment. Other avenues include the addition of automatic recommendation of substitution matrices so that the researcher can quickly narrow the number of matrices to be evaluated and the incorporation of visual alignment clustering to make comparison more intuitive to users.

## AVAILABILITY OF DATA AND MATERIALS

**Project name:** SubVis

**Project home page - Package:** https://cran.r-project.org/web/packages/SubVis/

**Project home page - Demo videos:** https://github.com/sabarlowe/SubVis

**Operating system(s):** Platform independent

**Tested browsers:** Mozilla Firefox and Google Chrome

**Programming languages:** R and JavaScript

**Other requirements:** R (> 3.3.0), Shiny (R package), Biostrings (R package), and a web browser

**License:** GNU GPL > 3

**Data:** The FASTA sequences for *G-protein coupled receptor 6 isoform b* (NP_005275.1), *G-protein coupled receptor 12* (NP_005279.1), DDX4 *Homo sapiens* (AAH47455.1), DDX4 *Xenopus laevis* (NP_001081728.1), and supplemental sequences were downloaded from the Protein database at the National Center for Biotechnology Information (*NCBI Resource Coordinators, 2016*). The data used in the manuscript, supplemental sequences, and supplemental custom matrices are provided as part of the software package.

### Funding

Robert T. Youker is supported by the American Heart Association (12SDG8960000). The funders had no role in study design, data collection and analysis, decision to publish, or preparation of the manuscript.

### Grant Disclosures

The following grant information was disclosed by the authors:
American Heart Association: 12SDG8960000.

### Competing Interests

The authors declare there are no competing interests.

### Author Contributions

- Scott Barlowe conceived and designed the experiments, performed the experiments, analyzed the data, contributed reagents/materials/analysis tools, wrote the paper, prepared figures and/or tables, and reviewed drafts of the paper.
- Heather B. Coan and Robert T. Youker conceived and designed the experiments, wrote the paper, reviewed drafts of the paper, and provided expert feedback.

### Data Availability

  Source Code: The Comprehensive R Archive Network:
  https://cran.r-project.org/web/packages/SubVis/index.html.
  Demo Videos (Part 1 and Part 2): https://github.com/sabarlowe/SubVis.

## Supplemental Information

Supplemental information for this article can be found online at http://dx.doi.org/10.7717/peerj.3492#supplemental-information.

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
