# Peer review of "SubVis: an interactive R package for exploring the effects of multiple substitution matrices on pairwise sequence alignment"

_PeerJ, doi:10.7717/peerj.3492_

## Round 0.1 · original submission · Major Revisions

Both reviewers have a number of suggestions that will improve your manuscript.

Reviewer 1 ·

Basic reporting

No comment.

Experimental design

No comment.

Validity of the findings

No comment.

Additional comments

The authors present an R package for visualizing pairwise amino-acid sequence alignments under a variety of substitution matrices, with varying penalty parameters. Overall, I think that this will be a useful tool, especially if/when the authors can extend SubVis to multiple sequence alignments. I have only minor comments:

1) The authors leave out a substantial body of literature on assessing alignment quality. I recommend the authors perform a more thorough literature search. It might be useful to start with some of these papers, and references within, to better introduce the field of alignment uncertainty:
- https://doi.org/10.1093/nar/gkv318
- https://www.ncbi.nlm.nih.gov/pubmed/22407710
- https://www.ncbi.nlm.nih.gov/pubmed/18218900

2) Can the authors clarify their "alignment algorithm" used in SubVis? Given that alignment is pairwise in SubVis, this should be straightforward to address. The authors could consider, if relevant, comparing their implementation to those of popular alignment platforms like Clustal, Mafft, Muscle, etc.

3) The authors may wish to distinguish between substitution matrices as used for sequence alignments (PAM, BLOSUM) and substitution matrices used as *phylogenetic* models of sequence evolution, such as JTT, WAG, LG, etc (see here: https://doi.org/10.1093/oxfordjournals.molbev.a003851) . For example, on pg 2 line 82-83, the authors write "There are few tools for addressing the complex problem of choosing the most appropriate substitution matrix for a given set of proteins." For phylogenetics, in fact, procedures or selecting an appropriate substitution matrix are fairly standardized and well-examined. This may not be the case for sequence alignment, as the authors note with this paper.

4) The authors spend a fair bit of text justifying their use of R and Shiny. Because these are such common/standard tools in the bioinformatics field, the authors could consider (at their discretion) reducing this repetition.

5) The phrasing within SubVis of "pattern" and "subject" is a bit confusing and could be replaced with "sequence 1" and "sequence 2" for an improved user experience. A single location to upload both sequences, rather than two text boxes, would also be useful (this is just a recommendation). Similarly, I would strongly recommend adding explicit usage into the Vignette (rather than just descriptive), e.g.:
> library(SubVis)
> SubVis::startSubVis() # or simply startSubVis()

The vignette could also be improved on CRAN with RMarkdown rendering (currently rendered fine in the SubVis help, but not for CRAN, where many users will start off).

6) The authors must include information on what PID<1,2,3,4> mean within SubVis, rather than just in the manuscript. Even in the Vignette would be helpful, but this needs to be somewhere. Most users will not know a priori what PIDs refer to.

7) Typo in caption for Table 2, where the citation Pommie ́ et al. is given twice. Perhaps missing parenthesis?

Reviewer 2 ·

Basic reporting

The manuscript is written in clear, professional English, and well organized.
However, sentence structure and word choice can be improved.
The manuscript provides sufficient background and follows a clear structure.

The manuscript repeats many statement throughout. It could be condensed and made more concise.

All data required to reproduce the case study was provided.

The presented figures are mostly relevant. Figure 1-3 do not provide any relevant additional information and only inflate the length of the manuscript.

Overall the work appears valid, but revisions are necessary.

Experimental design

No comment

Validity of the findings

Comparing statements made in Line 100 and Line 126, It appears that the work claims to address issues and provide solutions it does not deliver on. The authors should consider to provide a more clear statement how their tool differs from other available tools.

Referring to the view shown in figure 5: The histogram with the amino acid count per column does not represent the amino acid count at each position of the alignment as claimed but the amino acid count in that column. This is an important difference as the same column can represent different positions in the different alignments due to gaps.

Additional comments

Line 12: “Why proteins mutate” is out of the scope of this work and can not be determined by alignment tools.
Line 13: "Mutations can occur when ... is substituted for another" <- This is a mutation, no can. But is it necessary to define what a mutation is?
Line 15: Alignment quality does not provide reliable insight into sequence relatedness. Distantly related sequences can be highly conserved when under strong selection. Similar, Closely related sequences can have low similarity when under diverged or negative frequency selection.
Line 31-33: The abstract is not the place to advertise future work.
Line 39: Colloquial word choice, consider replacing “clues” by “insights”
Line 39: “Introduced purposefully”; Do the authors refer to mutations under selection or a higher power i.e. god?
Equation 1: Please explain what lambda is.
Line 48: “target frequencies” consider “observed frequencies”
Line 49-50: “conservative and non-conservative” used while conserved is used in Line 249-250. Both descriptions seem to describe the same type of mismatch. From the context, I think in both cases conservative should be used
Line 63: No reasons are provided as to why certain matrices may be inadequate for a certain task. Please provide a brief comment.
Line 77-78: "After choosing a particular..." Sentence structure should be improved
Line 83-84: "Bulka et al. builds on the work... " Sentence structure should be improved
Line 84: "... builds on work of Nakai et al. (Nakai et al. 1988) ..." cite as "... builds on work of Nakai et al. (1988) ..."
Line 167-168: "The intersection of the ..." consider "Each element (or entry) of the matrix represents ...". However, I am not convinced this sentence is necessary at all.
Line 168: “scoring values consistent with pairwiseAlignment function”; please provide details what that means. Is the scoring different from what was established in the introduction. If not, why talk about it?
Line 188: “communicates user updates to the visualization component”, Consider “communicates progress to the visualization component”
Line 307-311: Sentences are unpleasant to read, please restructure.
Line 307-309: The authors seem to try to make a point here that is unclear to me. The used obviously differ. Identification of differences in substitution scores can directly be obtained from the matrices and is not specific to alignment regions. Please clarify.
Line 309: The properties of the amino acids are classified, not the properties of the alignment.
Line 326,328: at CRAN, at GitHub, “on” is more appropriate
Line 328: The authors should not cite their supplementary material but rather refer to it.

Citing Pagel et al. 2016 multiple time for the same statement.
Citing Chang et al. 2016 multiple time for the same same statement.
Citing R Core Team 2013 multiple time for the same same statement.

---

## Round 0.2 · Minor Revisions

Both reviewers are satisfied with your revision in principle but have identified a few minor remaining issues. Please address those. Once you have addressed them, I will formally accept your manuscript. I don't expect that there will be another round of review.

Reviewer 1 ·

Basic reporting

-

Experimental design

-

Validity of the findings

-

Additional comments

The authors have done a nice job improving their manuscript. I have one very minor comment, which I expect can be dealt with during proofing. The authors now include the sentence (pg 2 lines 71-3):
"There are several standardized matrices for phylogenetic inference models, such as JTT (Jones et al., 1992) and WAG (Whelan and Goldman, 2001), but these matrices rely on a single set of stationary frequencies to describe protein family evolution."

This is not true. Most commonly, these models are used by taking the inferred empirical exchangeability matrix but also taking *empirical* frequencies specific to the dataset at hand, rather than using the model's a priori frequencies. As such, I recommend that the authors amend the sentence as follows: "There are several standardized matrices for phylogenetic inference models, such as JTT (Jones et al., 1992) and WAG (Whelan and Goldman, 2001), but these matrices **were not developed to be used in sequence alignment**". Essentially, replace "but these matrices rely on a single set of stationary frequencies to describe protein family evolution" with something that says that the matrices are not technically built for alignment but only for phylogeny.

Reviewer 2 ·

Basic reporting

The Authors were able to make significant improvements to the manuscript and address all relevant points.

I think the Manuscript is overall ready for acceptance. I have only minor comments that would further improve the manuscript. However, I don't think my comments are preventive for the acceptance of this work.

- Line 70-79: Here, I stand in opposition to my colleague reviewing this paper as well. I find this section awkward and forced. While Alignments are at the core of every phylogenetic study, they are usually not performed on the amino acid level but on the nucleotide and codon level. Both are not subject of this paper and the tool. If phylogenetics is supposed to be subject of this paper as well, it has to be introduced more carefully.

- Line 102-106: I suggest to refrain from using a list form to transition out of the introduction.

- Line 130: I don't think claims should be made without any kind of short description as to why it is the case. I don't think this sentence is necessary. If left, I suggest adding another sentence with a brief explanation as to why these methods are flawed.

- Line 147: I suggest deleting "sometimes problematic" as the problem with summery statistics is covered earlier in the text.

- Line 151: "of a substitution matrix" instead of "to a"?

Experimental design

no comment

Validity of the findings

no comment

---

## Round 0.3 · accepted · Accept

Thank you for making these final revisions.